# Serum 25(OH)D levels and mortality risk among middle-aged and elderly populations in the U.S.: A prospective cohort study

Yaowen Hu[1☉], Faliang Gao[2☉], Yuan Yang[3], Wei Yang[3], Huibo He[3], Jie Zhou[3], Yujie Zhao[1], Xi Chen[3]*, Wenyan Zhao[3]*, Xiaopeng He[ID][4]*

**1** Center for General Practice Medicine, Healthcare Department, Zhejiang Provincial People's Hospital (Affiliated People's Hospital, Hangzhou Medical College), Hangzhou, Zhejiang, China, **2** Center for Rehabilitation Medicine, Department of Neurosurgery, Zhejiang Provincial People's Hospital (Affiliated People's Hospital, Hangzhou Medical College), Hangzhou, Zhejiang, China, **3** Center for General Practice Medicine, General Practice and Health Management Center, Zhejiang Provincial People's Hospital (Affiliated People's Hospital, Hangzhou Medical College), Hangzhou, Zhejiang, China, **4** Center for General Practice Medicine, Department of Party and Administrative Affairs Office, Zhejiang Provincial People's Hospital (Affiliated People's Hospital, Hangzhou Medical College), Hangzhou, Zhejiang, China

☉ These authors contributed equally to this work.
* hexiaopeng411@163.com (XPH); zhaowenyan@hmc.edu.cn (WYZ); vipxichen@126.com (XC)

## Abstract

### Objective

To investigate the prevalence of vitamin D deficiency and its relationship with all-cause and cause-specific mortality among middle-aged and elderly populations in the U.S.

### Methods

Data were sourced from the National Health and Nutrition Examination Survey (NHANES) 2001–2018. A total of 22,130 participants aged 40–70 years were included. Serum 25-hydroxy vitamin D [25(OH)D] concentrations were measured and categorized. The primary outcome was all-cause mortality, and secondary outcomes were cardiovascular disease (CVD) and cancer mortality. Multivariable-adjusted models and various statistical analyses were employed.

### Results

The prevalence of vitamin D deficiency (≤50.00 nmol/L) was 33.59%, and insufficiency (≤75.00 nmol/L) was 71.74%. For all-cause mortality, the multivariate adjusted hazard ratios (HRs) across different 25(OH)D levels (< 25.00, 25.00–49.99, 50.00–74.99, and ≥ 75.00 nmol/L) were 1.00, 0.78 (0.65, 0.93) $p = 0.0069$, 0.59 (0.49,0.72) $p < 0.0001$, and 0.54 (0.44, 0.66) $p < 0.0001$ respectively. Similar patterns were observed for CVD mortality. There was no significant difference in cancer mortality

**Data availability statement:** All relevant data are within the manuscript and its Supporting Information files.

**Funding:** This study were financially supported by Zhejiang Province Health Science and Technology Project (grant no. 2024KY652) and Zhejiang Province Traditional Chinese Medicine Science and Technology Project (grant no. 2025ZL188). The funders had no role in study design, data collection and analysis, decision to publish, or preparation of the manuscript.

**Competing interests:** The authors have declared that no competing interests exist.

**Abbreviations:** 25(OH)D, 25-Hydroxyvitamin D; CVD, Cardiovascular Disease; NHANES, National Health and Nutrition Examination Survey; HR, Hazard Ratio; CI, Confidence Interval; BMI, Body Mass Index; PIR, Poverty Income Ratio; RAAS, Renin-Angiotensin-Aldosterone System; PTH, Parathyroid Hormone; LC-MS/MS, Liquid Chromatography-Tandem Mass Spectrometry; NCEH, National Center for Environmental Health; CDC, Centers for Disease Control and Prevention; RA, Rheumatoid Arthritis; VDREs, Vitamin D Response Elements; VDR, Vitamin D Receptor; TDECs, Tumor-derived endothelial cells; VEGF, Vascular endothelial growth factor

between the moderately deficient and severely deficient groups, but lower mortality was found in the insufficient and sufficient groups compared to the severely deficient group. An L-shaped association between serum vitamin D levels and mortality was identified. Subgroup analyses were consistent with the main findings.

## Conclusion

This study found that higher serum 25-hydroxyvitamin D concentrations are linked to lower all-cause, cardiovascular, and cancer mortality. The relationship is non-linear: increases in concentration reduce death risk below a certain threshold, but above it, the association weakens. Further research is needed to understand causal mechanisms.

---

## Introduction

Vitamin D, a unique micronutrient with endocrine functions, serves dual roles in human physiology as both an essential nutrient and a steroid hormone. Its primary biological action involves regulation of calcium-phosphate homeostasis through genomic and non-genomic pathways [1], while pleiotropic effects extend to modulation of lipoprotein metabolism, bone remodeling, cardiovascular function, and immune regulation [2–5]. Mechanistically, vitamin D promotes epidermal cell proliferation/differentiation, mitigates oxidative stress and inflammatory responses, suppresses parathyroid hormone (PTH) secretion, and regulates the renin-angiotensin-aldosterone system (RAAS) [2–5]. Global epidemiological data reveal a pandemic of vitamin D insufficiency, with particular clinical significance given its association with chronic disease pathogenesis [6]. Observational studies consistently demonstrate inverse correlations between serum 25-hydroxyvitamin D [25(OH)D] concentrations and disease burden, including cardiovascular mortality [7], osteoporotic fractures [8], autoimmune disorders [9–11], malignancies, and respiratory infections [6]. In bone metabolism, vitamin D critically maintains mineralization balance by enhancing intestinal calcium absorption and regulating bone turnover rates [8]. Cardiovascular implications extend to modulation of myocardial function, with deficiency states independently predicting incident heart failure and atherosclerotic events [7]. The aging population demonstrates heightened vulnerability to hypovitaminosis D due to multisystem decline: impaired dermal synthesis (reduced 7-dehydrocholesterol levels), diminished renal 1α-hydroxylase activity, and decreased outdoor mobility collectively exacerbate deficiency risk [12]. U.S. national surveys indicate 5.9% prevalence of severe deficiency (25(OH)D < 30 nmol/L [12 ng/mL]) and 24% prevalence of suboptimal levels (<50 nmol/L [20 ng/mL]) among adults [13], underscoring the need for targeted assessment in older demographics. This population-based study aims to quantify the prevalence of vitamin D deficiency among middle-aged and older U.S. adults, and evaluate longitudinal associations between 25(OH)D status and all-cause mortality risk, with particular focus on dose-response relationships and threshold effects.

## Method

### Study design

The data for this study were sourced from the National Health and Nutrition Examination Survey (NHANES), a public health program conducted by the National Center for Health Statistics (NCHS). Established in the early 1960s, NHANES collects demographic and health data through questionnaires, physical exams, and laboratory tests administered at mobile health screening centers (available at: https://wwwn.cdc.gov/nchs/nhanes/Default.aspx). The survey is conducted biennially and data are released every two years. The study utilized data from nine consecutive 2-year survey cycles (2001–2002, 2003–2004,..., 2017–2018) to conduct prospective secondary analyses, linking NHANES data with mortality records from the National Death Index. All NHANES protocols were approved by the ethics review board of the National Center for Health Statistics, and participants provided written informed consent, outlining their awareness of the study's aims, procedures, and potential risks, with their voluntary participation being the basis for their involvement.Written informed consent was obtained from all study participants. Additional protocol details are accessible through the designated resources on the CDC's official NHANES portal (https://www.cdc.gov/nchs/nhanes/index.htm).

### Study population

This study utilized a retrospective cohort design, analyzing data from nine consecutive 2-year cycles (spanning 2001–2002–2017–2018) of the National Health and Nutrition Examination Survey (NHANES), which included a total of 91,352 participants. To minimize potential survival bias in our analysis, we specifically focused on middle-aged and older adults aged 40–70 years, resulting in a cohort of 24,580 individuals. After excluding participants without valid vitamin D level measurements (n = 2,407) and those without complete all-cause mortality records (n = 43), the final analytic sample consisted of 22,130 eligible participants (Fig 1).

### Exposure

The serum 25(OH)D concentration measurements from the National Health and Nutrition Examination Survey (NHANES) were initially collected using the DiaSorin RIA kit (Stillwater, MN) between 2001 and 2006. However, starting in 2007, the study transitioned to measuring serum 25(OH)D concentrations using a standardized liquid chromatography-tandem mass spectrometry (LC-MS/MS) method. To ensure comparability between the older DiaSorin RIA-based measurements (2001–2006) and the newer LC-MS/MS method, the CDC adjusted the 2001–2006 data to align with the LC-MS/MS measurement scale. The experimental serum 25(OH)D concentration data were therefore converted to equivalent 25(OH)D measurements using regression analysis to match the standardized LC-MS/MS method. Detailed methodology information is available on the NHANES website [14]. Additionally, according to the Clinical Laboratory Guidelines in Endocrinology, serum 25(OH)D concentrations were categorized into four groups: severely deficient (<25.0 nmol/L), moderately deficient (25.0–49.9 nmol/L), insufficient (50.0–74.9 nmol/L), and sufficient (≥75.0 nmol/L) [15].

### Outcome

The primary endpoint of this study was all-cause mortality, with secondary outcomes including cause-specific mortalities specifically attributed to cardiovascular diseases (CVD) and cancer. Mortality data were sourced from death records publicly accessible through the National Center for Health Statistics (NCHS). Follow-up records were sourced from the National Health and Nutrition Examination Survey (NHANES) up to the date of death on December 31, 2019. Causes of death were determined by NCHS based on the International Classification of Diseases, 10th Revision (ICD-10). CVD mortality was defined as deaths resulting from cardiovascular diseases, including heart disease (specific ICD-10 codes I00-I09, I11, I13, I20-I51) and cerebrovascular diseases (specific ICD-10 codes I60-I69). Cancer mortality was defined as deaths due to malignant neoplasms (specific ICD-10 codes C00-C97).

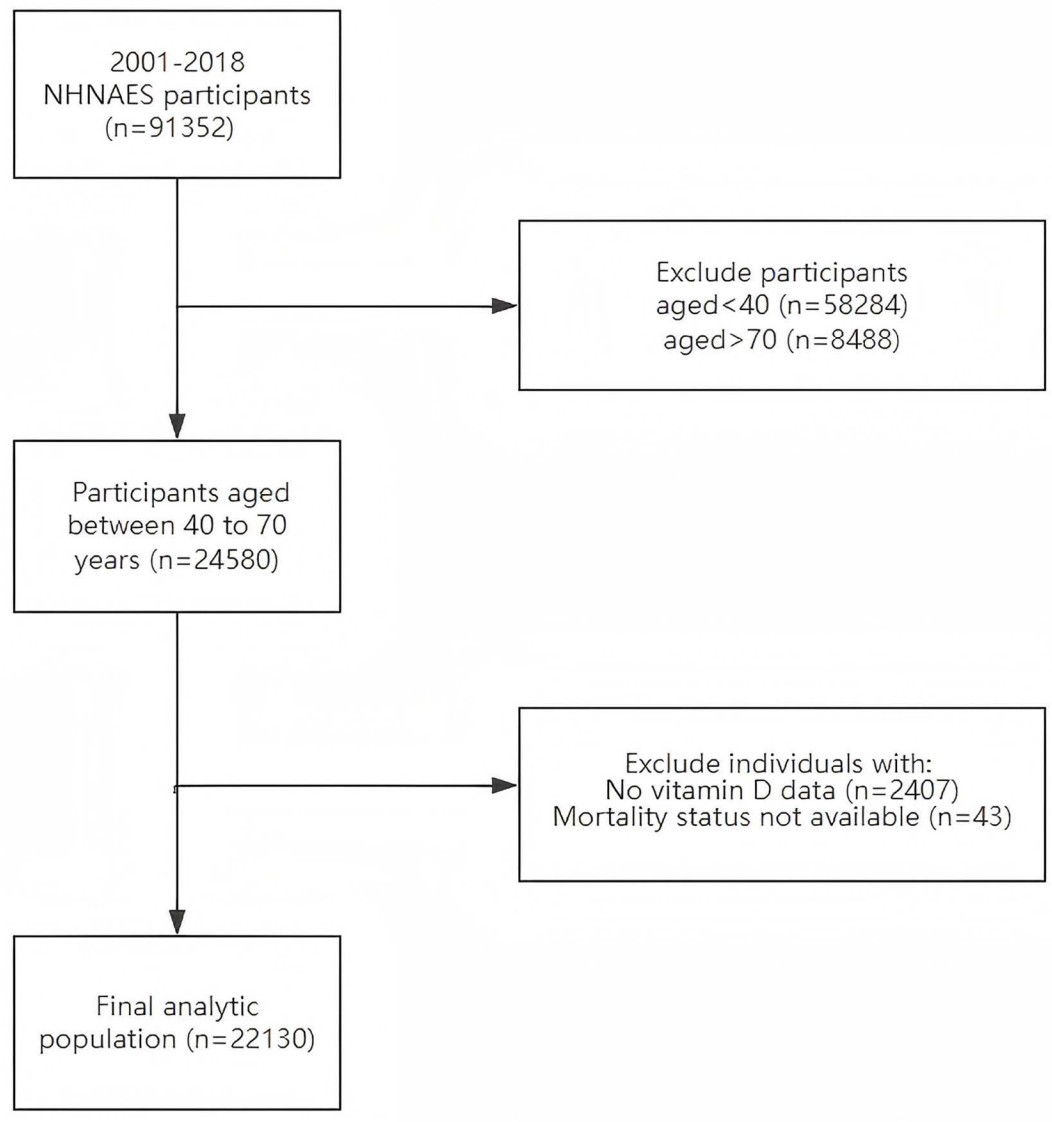

**Fig 1. Flow chart of participants.**

## Covariates

The race/ethnicity of participants was categorised as Mexican American, Other Hispanic, Non-Hispanic White,Non-Hispanic Black and other race. The level of education was categorised as less than high school, high school or equivalent and college or above. Marital status was categorised as non-single (including married or with a partner) and single (including widowed, divorced, separated or never married). The assessment of physical activity was conducted using the Global Physical Activity Questionnaire (GPAQ) developed by the World Health Organization (WHO) [16]. The weekly amount of physical activity was calculated as metabolic equivalent minutes (MET minutes), with physical activity ≥ 600 MET minutes/week considered sufficient and physical activity<600 MET minutes/week considered insufficient [16]. Family income-poverty ratio was categorised less than 1.3, between 1.3 to 3.5 and more than 3.5. Alcohol consumption was categorised as Never, Low-to-moderate drinker (defined as, <2 drinks/day in the past 12 month), or heavy drinker (defined

as ≥2 drinks/day in the past 12 month). Smoking grouped by never smoked, current smoker, former smoker. Body mass index (BMI) is divided into three categories: <25 (normal), ≥25 to <30 (overweight), and ≥30 kg/m2 (obese), and is based on weight (kg) divided by the square of height (m). Calorie intake data are derived from participants' 24-hour food and beverage consumption recalled during interviews (from midnight to midnight). The presence of hypertension, diabetes, stroke, and cardiovascular disease (CVD) was determined based on whether the individual had been informed of their condition by a medical professional. Hypertension, cardiovascular disease (CVD) and stroke were categorised as either "Yes" or "No," while diabetes was categorised as "Yes," "No," or "Borderline".

## Statistical analyses

Continuous variables that follow a normal distribution were described as mean values accompanied by their standard deviation (SD), while non-normally distributed continuous variables were expressed as median (quartile 1, quartile 3), and the Mann-Whitney U-test was employed for comparisons. Categorical variables were presented as numbers (%), and chi-square tests were used to assess differences. To evaluate the association between serum 25(OH)D levels and mortality risk, multivariable-adjusted models and smooth curve fitting were applied. Three models were run concurrently, adhering to the STROBE guidelines: Model 1 served as the unadjusted model, Model 2 adjusted for age, sex, and race, and Model 3 adjusted for all covariates listed in Table 1. The Cox proportional hazards regression model was utilized to calculate hazard ratios (HR) along with 95% confidence intervals (95% CI) across the three models. Sensitivity analyses were conducted to reassess the relationship between serum 25(OH)D levels and mortality by treating 25(OH)D levels as a categorical variable. To test the robustness of the findings, subgroup analyses were performed according to sex, BMI, lifestyle (smoking status, alcohol consumption), history of chronic disease(hypertension and diabetes and CVD) using stratified Cox proportional hazards models.Cumulative survival rate analysis was executed using Kaplan–Meier curves and log-rank statistics, comparing different groups based on their serum 25(OH)D levels. All tests conducted were two-tailed, with statistical significance defined as a p value less than 0.05. All data analyses were carried out using the R statistical software package (http://www.R-project.org, The R Foundation for Statistical Computing), EmpowerStats (http://www.empowerstats.com, X&Y Solution, Inc).

## Results

### Baseline characteristics of study participants

Among 22,130 participants aged between 40 and 70 years included in this study, baseline characteristics were comprehensively documented (Table 1). The mean age of the participants was 54.6 ± 8.9 years. The mean serum 25(OH)D concentration was 62.9 ± 26.5 nmol/L. Among the participants, 33.6% were vitamin D deficient (≤50.00 nmol/L), while 71.7% were vitamin D insufficient (≤75.00 nmol/L), indicating a significant proportion of individuals with sub-optimal vitamin D levels. Table 1 provides detailed baseline demographics of the screened population based on serum 25(OH)D levels. Demographically, individuals with higher 25(OH)D concentrations were more likely to be non-Hispanic white, non-single, with higher levels of education and household income, lower body mass index (BMI), and less likelihood of being smokers and drinkers. Notably, individuals in the severely deficient group exhibited a significantly higher likelihood of developing hypertension, diabetes, and stroke compared to those with moderate deficiency, insufficient levels, or sufficient 25(OH)D levels.

### Association of 25(OH)D concentration with all-cause and cause-specific mortality

During a median follow-up of 103.0months (interquartile range: 57.0–154.0months), a total of there were 2345 all-cause deaths, including 636 deaths related to cardiovascular disease and 684 cancer-related deaths (Table 2). To explore the association between serum 25(OH)D levels and mortality, three Cox regression models were constructed. Multivariate

**Table 1. Baseline characteristics of middle-aged and older adult by serum 25(OH)D concentration in NHANES 2001-2018.**

| | Severely deficient | Moderately deficient | Insufficient | Sufficient | p value |
|---|---|---|---|---|---|
| **Vitamin D level (nmol/L)** | 20.29±3.62 | 38.90±7.02 | 62.24±6.97 | 95.19±21.17 | <0.001 |
| **Vitamin D level group (nmol/L)** | <25 | 25-49.9 | 50-74.9 | ≥75 | |
| **Number of participants** | 992 | 6443 | 8442 | 6253 | |
| **Age(Year)** | 53.56±8.51 | 53.86±8.85 | 54.22±8.92 | 55.89±8.82 | <0.001 |
| **BMI** | 32.60±9.33 | 30.93±7.29 | 29.59±6.31 | 28.43±6.30 | <0.001 |
| **Caloric intake (kcal)** | 1779.00 (1314.00-2346.00) | 1874.00 (1376.00-2524.00) | 1957.00 (1449.50-2607.00) | 1927.50 (1438.00-2552.00) | <0.001 |
| **BMI group** | | | | | <0.001 |
| <25 | 181 (18.85%) | 1239 (19.55%) | 1878 (22.49%) | 1948 (31.40%) | |
| 25-29.9 | 265 (27.60%) | 1990 (31.40%) | 3096 (37.07%) | 2224 (35.85%) | |
| ≥30 | 514 (53.54%) | 3109 (49.05%) | 3377 (40.44%) | 2032 (32.75%) | |
| **Gender** | | | | | <0.001 |
| Male | 405 (40.79%) | 3138 (48.63%) | 4471 (52.84%) | 2825 (45.09%) | |
| Female | 588 (59.21%) | 3315 (51.37%) | 3991 (47.16%) | 3440 (54.91%) | |
| **Race** | | | | | <0.001 |
| Mexican American | 140 (14.10%) | 1428 (22.13%) | 1671 (19.75%) | 576 (9.19%) | |
| Other Hispanic | 58 (5.84%) | 550 (8.52%) | 956 (11.30%) | 502 (8.01%) | |
| Non-Hispanic White | 129 (12.99%) | 1599 (24.78%) | 3632 (42.92%) | 3732 (59.57%) | |
| Non-Hispanic Black | 600 (60.42%) | 2255 (34.94%) | 1360 (16.07%) | 762 (12.16%) | |
| Other race | 66 (6.65%) | 621 (9.62%) | 843 (9.96%) | 693 (11.06%) | |
| **Education** | | | | | <0.001 |
| less than high school | 286 (28.80%) | 2020 (31.30%) | 2365 (27.95%) | 1174 (18.74%) | |
| high school or | 241 (24.27%) | 1507 (23.35%) | 1841 (21.76%) | 1427 (22.78%) | |
| equivalent | 466 (46.93%) | 2916 (45.19%) | 4252 (50.25%) | 3660 (58.42%) | |
| college or above | | | | | |
| **Martial Status** | | | | | <0.001 |
| Non-single | 488 (49.14%) | 3942 (61.09%) | 5836 (68.97%) | 4315 (68.87%) | |
| Single | 504 (50.76%) | 2507 (38.85%) | 2617 (30.93%) | 1947 (31.08%) | |
| **Physical activity** | | | | | <0.001 |
| insufficient | 199 (20.06%) | 1648 (25.58%) | 2433 (28.82%) | 2018 (32.27%) | |
| sufficient | 116 (11.69%) | 992 (15.40%) | 1762 (20.87%) | 1441 (23.04%) | |
| **PIR[a]** | | | | | <0.001 |
| <1.3 | 332 (36.09%) | 1959 (33.36%) | 2166 (27.91%) | 1317 (22.84%) | |
| 1.3-3.5 | 371 (40.33%) | 2217 (37.76%) | 2783 (35.86%) | 1809 (31.37%) | |
| ≥3.5 | 217 (23.59%) | 1696 (28.88%) | 2812 (36.23%) | 2640 (45.79%) | |
| **Smoking** | | | | | <0.001 |
| Never | 458 (46.12%) | 3208 (49.71%) | 4417 (52.20%) | 3232 (51.59%) | |
| Former | 178 (17.93%) | 1525 (23.63%) | 2383 (28.16%) | 1876 (29.94%) | |
| current | 356 (35.85%) | 1709 (26.48%) | 1660 (19.62%) | 1155 (18.44%) | |
| **Alcohol consumption** | | | | | <0.001 |
| Never | 205 (20.64%) | 1246 (19.31%) | 1470 (17.37%) | 929 (14.83%) | |
| Low-to-moderate drinker | 191 (19.23%) | 1314 (20.36%) | 2051 (24.24%) | 1700 (27.13%) | |
| Heavy drinker | 370 (37.26%) | 2315 (35.87%) | 3193 (37.73%) | 2335 (37.27%) | |

*(Continued)*

**Table 1.** (Continued)

| | Severely deficient | Moderately deficient | Insufficient | Sufficient | *p* value |
|---|---|---|---|---|---|
| **Hypertension** | | | | | <0.001 |
| Yes | 498 (50.20%) | 2840 (44.11%) | 3318 (39.26%) | 2674 (42.76%) | |
| No | 494 (49.80%) | 3599 (55.89%) | 5133 (60.74%) | 3579 (57.24%) | |
| **Diabetes** | | | | | <0.001 |
| Yes | 216 (21.75%) | 1189 (18.43%) | 1259 (14.89%) | 903 (14.42%) | |
| No | 753 (75.83%) | 5060 (78.45%) | 6994 (82.74%) | 5173 (82.60%) | |
| Borderline | 24 (2.42%) | 201 (3.12%) | 200 (2.37%) | 187 (2.99%) | |
| **CVD** | | | | | 0.669 |
| Yes | 41 (4.17%) | 276 (4.29%) | 335 (3.97%) | 243 (3.89%) | |
| No | 943 (95.83%) | 6155 (95.71%) | 8096 (96.03%) | 6009 (96.11%) | |
| **Stroke** | | | | | <0.001 |
| Yes | 64 (6.45%) | 267 (4.14%) | 260 (3.08%) | 250 (4.00%) | |
| No | 929 (93.55%) | 6179 (95.86%) | 8190 (96.92%) | 6005 (96.00%) | |

ª Continuous variables that follow a normal distribution were presented as mean ± standard deviation, while non-normally distributed variables were expressed as median (quartile 1, quartile 3). Categorical variables were presented as numbers (%).

ᵇ Abbreviations:PIR, ratio of family income to poverty; BMI, body mass index; CVD, cardiovascular disease.

adjustments included several factors: age, sex, race, education, marital status, body mass index, alcohol consumption, smoking, cardiovascular disease, stroke, hypertension, diabetes mellitus, and family income-poverty ratio. For all-cause mortality, the multivariate-adjusted hazard ratios (HRs) and 95% confidence intervals (CIs) across increasing levels of serum 25(OH)D (<25.00, 25.00–49.99, 50.00–74.99, and ≥75.00 nmol/L) were 1.00 (reference), 0.78 (0.65–0.93), 0.59 (0.49–0.72), and 0.54 (0.44–0.66) (Model 3). Similarly, for cardiovascular disease (CVD) mortality, the HRs and 95% CIs were 1.00 (reference), 0.71 (0.51–0.99), 0.55 (0.39–0.78), and 0.53 (0.36–0.77) across the four categories of serum 25(OH)D levels (Model 3). Additionally, for cancer mortality, the HRs and 95% CIs were 1.00 (reference), 0.85 (0.60–1.20), 0.62 (0.43–0.89), and 0.63 (0.42–0.92) across the same categories (Model 3). Compared with the group with serum 25(OH)D < 25.00 nmol/L, middle-aged and older adults with higher levels of serum 25(OH)D (≥25.00 nmol/L) demonstrated significant associations with lower all-cause, CVD, and cancer mortality rates.

## Analyses of the dose–response relationship between 25(OH)D concentration and all-cause and cause-specific mortality

The association between serum vitamin D levels and all-cause and cause-specific mortality was evaluated using a generalized additive model with smooth curve fitting (restricted cubic spline method) on a continuous scale. The fully adjusted smooth curve fitting model indicated an L-shaped association between serum vitamin D levels and both all-cause and cause-specific mortality (Fig 2). We further investigated the relationship between the risk of all-cause and cause-specific (CVD and cancer) mortality and 25(OH)D levels. As shown in Table 3, a nonlinear relationship was observed between 25(OH)D levels and mortality risk. When below 54.80 nmol/L, an increase of one unit in 25(OH)D concentration was associated with a 1.9% reduction in all-cause mortality (HR 0.981; 95% CI: 0.976, 0.985; p < 0.0001). When 25(OH)D levels exceeded 54.80 nmol/L, there was no significant association with all-cause mortality (HR 0.999; 95% CI: 0.996, 1.002; p = 0.6017). The inflection points for cause-specific mortality (CVD and cancer) were 44.70 nmol/L and 58.70 nmol/L, respectively. Below these inflection points, a one-unit increase in 25(OH)D concentration was associated with a 2.9% reduction in mortality for CVD (HR 0.971; 95% CI: 0.959, 0.983; p < 0.0001) and a 1.8% reduction for cancer (HR 0.982; 95% CI: 0.975, 0.990; p < 0.0001). When 25(OH)D levels exceeded these inflection points, there was no significant

**Table 2. Hazard Ratios (95% CIs) for all-cause and cause-specific mortality according to serum 25(OH)D concentrations among participants with middle-aged and older adult in NHANES 2001–2018.**

**All-cause mortality**

| | Number | Model I | p value | Model II | p value | Model III | p value |
|---|---|---|---|---|---|---|---|
| Vitamin D level (nmol/L) | 2345 | 0.990 (0.988, 0.992) | <0.0001 | 0.986 (0.984, 0.989) | <0.0001 | 0.992 (0.990, 0.994) | <0.0001 |
| Severely deficient | 193 | ref | | ref | | ref | |
| Moderately deficient | 896 | 0.66 (0.56, 0.77) | <0.0001 | 0.60 (0.51, 0.70) | <0.0001 | 0.78 (0.65, 0.93) | 0.0067 |
| Insufficient | 793 | 0.45 (0.39, 0.53) | <0.0001 | 0.37 (0.32, 0.44) | <0.0001 | 0.59 (0.49, 0.72) | <0.0001 |
| Sufficient | 463 | 0.43 (0.36, 0.51) | <0.0001 | 0.33 (0.27, 0.39) | <0.0001 | 0.54 (0.44, 0.66) | <0.0001 |
| p for trend | | | <0.001 | | <0.001 | | <0.001 |

**CVD mortality**

| | Number | Model I | p value | Model II | p value | Model III | p value |
|---|---|---|---|---|---|---|---|
| Vitamin D level (nmol/L) | 636 | 0.988 (0.984, 0.991) | <0.0001 | 0.986 (0.982, 0.990) | <0.0001 | 0.992 (0.988, 0.997) | <0.0001 |
| Severely deficient | 65 | ref | | ref | | ref | |
| Moderately deficient | 246 | 0.53 (0.41, 0.70) | <0.0001 | 0.51 (0.38, 0.67) | <0.0001 | 0.71 (0.51, 0.99) | 0.0462 |
| Insufficient | 206 | 0.35 (0.27, 0.46) | <0.0001 | 0.32 (0.24, 0.43) | <0.0001 | 0.55 (0.39, 0.78) | 0.0008 |
| Sufficient | 119 | 0.33 (0.24, 0.45) | <0.0001 | 0.29 (0.21, 0.40) | <0.0001 | 0.53 (0.36, 0.77) | 0.0010 |
| p for trend | | | <0.001 | | <0.001 | | <0.001 |

**Cancer mortality**

| | Number | Model I | p value | Model II | p value | Model III | p value |
|---|---|---|---|---|---|---|---|
| Vitamin D level (nmol/L) | 685 | 0.995 (0.992, 0.998) | 0.0021 | 0.991 (0.988, 0.995) | <0.0001 | 0.995 (0.991, 0.999) | 0.0099 |
| Severely deficient | 43 | ref | | ref | | ref | |
| Moderately deficient | 254 | 0.83 (0.60, 1.15) | 0.2692 | 0.76 (0.55, 1.06) | 0.1051 | 0.85 (0.60, 1.20) | 0.3577 |
| Insufficient | 229 | 0.59 (0.42, 0.82) | 0.0014 | 0.49 (0.35, 0.68) | <0.0001 | 0.62 (0.43, 0.89) | 0.0089 |
| Sufficient | 158 | 0.66 (0.47, 0.92) | 0.015 | 0.50 (0.35, 0.71) | 0.0001 | 0.63 (0.43, 0.92) | 0.0177 |
| p for trend | | | <0.001 | | <0.001 | | <0.001 |

Model I: Non-adjusted. Model II: adjusted for age,sex and race/ethnicity.

Model III: adjusted for age, sex, race/ethnicity, education, martial Status, BMI, Caloric intake, Physical activity, alcohol consumption, smoking, comorbidities (stroke, hypertension and diabetes), family income-poverty ratio.

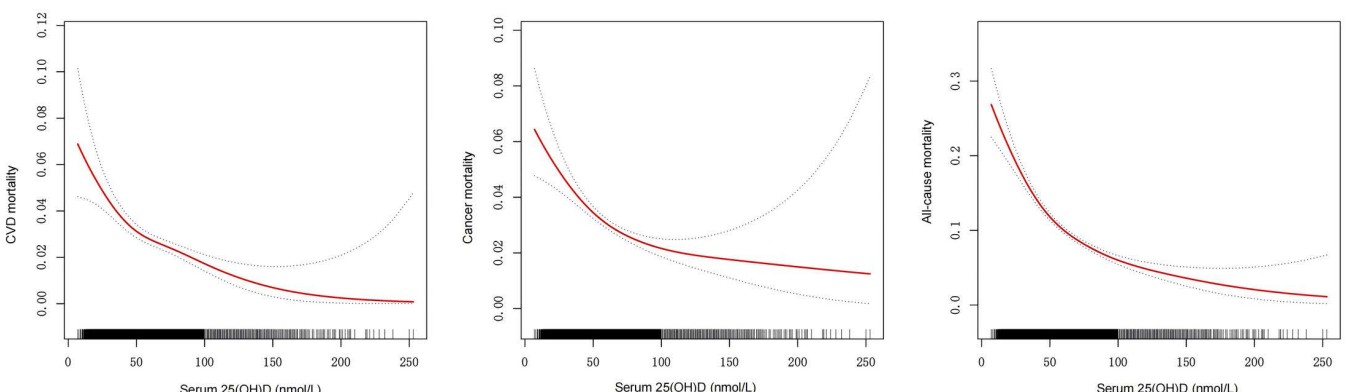

**Fig 2. Dose–response associations of serum 25(OH)D level with risk of All-cause (a), CVD (b), Cancer (c) mortality.** The red solid line represents the estimated risk of all-cause and cause-specific mortality, with blue dashed lines showing 95% CI, Analyses were adjusted for Gender, Race, Education, Martial Status, BMI Group, Drinking, Smoking, CVD, Caloric intake, Physical activity, Stroke, Hypertension, Diabetes, and PIR. PIR, ratio of family income to poverty; BMI, body mass index; CVD, cardiovascular disease.

**Table 3. Threshold effect analysis of serum 25(OH)D concentrations on all-cause and CVD mortality in middle-aged and elderly populations.**

| | Adjusted HR (95% CI)ᵃ, p-value | | |
|---|---|---|---|
| | Model I | Model II | Model III |
| **All-cause mortality** | | | |
| Fitting by the standard linear model | 0.990 (0.989, 0.992) <0.0001 | 0.988 (0.986, 0.990) <0.0001 | 0.992 (0.990, 0.994) <0.0001 |
| Inflection pointᵇ | 54.40 nmol/L | 55.20 nmol/L | 54.80 nmol/L |
| 25(OH)D concentrations<54.80 nmol/L | 0.975 (0.971, 0.979) <0.0001 | 0.972 (0.968, 0.976) <0.0001 | 0.981 (0.976, 0.985) <0.0001 |
| 25(OH)D concentrations≥54.80 nmol/L | 1.000 (0.998, 1.003) 0.8022 | 0.999 (0.996, 1.002) 0.5129 | 0.999 (0.996, 1.002) 0.6017 |
| p for Log-likelihood ratio | <0.001 | <0.001 | <0.001 |
| **CVD mortality** | | | |
| Fitting by the standard linear model | 0.988 (0.984, 0.991) <0.0001 | 0.988 (0.984, 0.992) <0.0001 | 0.993 (0.989, 0.997) 0.0005 |
| Inflection pointᵇ | 47.20 nmol/L | 46.90 nmol/L | 44.70 nmol/L |
| 25(OH)D concentrations<44.70 nmol/L | 0.963 (0.954, 0.972) <0.0001 | 0.962 (0.952, 0.972) <0.0001 | 0.971 (0.959, 0.983) <0.0001 |
| 25(OH)D concentrations≥44.70 nmol/L | 0.999 (0.994, 1.003) 0.5466 | 0.998 (0.993, 1.003) 0.4910 | 0.999 (0.994, 1.004) 0.7827 |
| p for Log-likelihood ratio | <0.001 | <0.001 | <0.001 |
| **Cancer mortality** | | | |
| Fitting by the standard linear model | 0.995 (0.992, 0.998) 0.0021 | 0.992 (0.989, 0.996) <0.0001 | 0.995 (0.991, 0.999) 0.0099 |
| Inflection pointᵇ | 58.60 nmol/L | 58.80 nmol/L | 58.70 nmol/L |
| 25(OH)D concentrations<58.70 nmol/L | 0.980 (0.974, 0.987) <0.0001 | 0.975 (0.968, 0.982) <0.0001 | 0.982 (0.975, 0.990) <0.0001 |
| 25(OH)D concentrations≥58.70 nmol/L | 1.005 (1.000, 1.010) 0.0307 | 1.004 (0.999, 1.009) 0.1073 | 1.004 (0.998, 1.009) 0.1798 |
| p for Log-likelihood ratio | <0.001 | <0.001 | <0.001 |

ᵃ Cox proportional hazards models were used to estimate hazard ratios (HRs) and 95% confidence intervals (95% CIs)

ᵇ Fitting model by two-piecewise Cox proportional hazards model

Model I: Non-adjusted.

Model II: adjusted for age,sex and race/ethnicity, education, martial Status.

Model III: adjusted for age, sex, race/ethnicity, education, martial Status, BMI, Caloric intake, Physical activity, alcohol consumption, smoking, comorbidities (stroke, hypertension and diabetes), family income-poverty ratio.

association with CVD mortality (HR 0.999; 95% CI: 0.994, 1.004; p = 0.7827) or cancer mortality (HR 1.004; 95% CI: 0.998, 1.009; p = 0.1798).

## Subgroup analyses

The subgroup analysis results examining the association between serum 25(OH)D and all-cause mortality are presented in Table 4. The findings from the stratified analysis of serum 25(OH)D are consistent with those from the multi-variable Cox regression analysis.

## Survival analyses

The Kaplan–Meier analysis revealed (Figs 3 and 4) that participants with vitamin D severely deficient group had a significantly lower survival probability compared to those with moderately deficiency, insufficiency, and sufficiency of serum vitamin D (both p < 0.001)

## Sensitivity analyses

We explored the potential for unmeasured confounding between serum 25(OH)D levels and all-cause and cause-specific mortality by calculating E-values. The E-value quantifies the magnitude of an unmeasured confounder that would be needed to negate the observed association between serum 25(OH)D and all-cause and cause-specific mortality [17].

**Table 4.  Association between serum 25(OH)D and all-cause mortality according to subgroup.**

| Characteristic | No. of participants | All-cause HR (95% CI) p value | CVD HR (95% CI) p value | Cancer HR (95% CI) p value |
|---|---|---|---|---|
| **Gender** | | | | |
| Male | 10814 | 0.77 (0.72, 0.82) <0.0001 | 0.71 (0.63, 0.80) <0.0001 | 0.90 (0.80, 1.02) 0.1015 |
| Female | 11316 | 0.73 (0.68, 0.79) <0.0001 | 0.67 (0.58, 0.77) <0.0001 | 0.78 (0.68, 0.89) 0.0003 |
| **Race** | | | | |
| Mexican American | 3812 | 0.77 (0.66, 0.89) 0.0004 | 0.66 (0.49, 0.89) 0.0065 | 0.93 (0.71, 1.22) 0.5961 |
| Other Hispanic | 2061 | 0.76 (0.60, 0.95) 0.0164 | 0.92 (0.59, 1.43) 0.7074 | 1.16 (0.78, 1.74) 0.4664 |
| Non-Hispanic White | 9079 | 0.69 (0.64, 0.74) <0.0001 | 0.67 (0.58, 0.78) <0.0001 | 0.73 (0.64, 0.83) <0.0001 |
| Non-Hispanic Black | 4967 | 0.84 (0.76, 0.93) 0.0004 | 0.81 (0.69, 0.96) 0.0159 | 0.93 (0.77, 1.11) 0.4091 |
| Other race | 2211 | 0.69 (0.53, 0.89) 0.0041 | 0.85 (0.52, 1.38) 0.5095 | 0.82 (0.50, 1.37) 0.4582 |
| **Education** | | | | |
| Less than high school | 5840 | 0.79 (0.73, 0.86) <0.0001 | 0.76 (0.65, 0.88) 0.0004 | 0.92 (0.78, 1.08) 0.3037 |
| High school or equivalent | 5007 | 0.81 (0.74, 0.89) <0.0001 | 0.67 (0.56, 0.81) <0.0001 | 0.93 (0.77, 1.11) 0.4052 |
| College or above | 11266 | 0.76 (0.71, 0.82) <0.0001 | 0.76 (0.65, 0.88) 0.0004 | 0.82 (0.72, 0.94) 0.0046 |
| **Martial Status** | | | | |
| Non-single | 14545 | 0.77 (0.72, 0.83) <0.0001 | 0.73 (0.64, 0.84) <0.0001 | 0.92 (0.82, 1.04) 0.1932 |
| Single | 7569 | 0.81 (0.76, 0.87) <0.0001 | 0.75 (0.66, 0.85) <0.0001 | 0.83 (0.73, 0.95) 0.0060 |
| **Physical activity** | | | | |
| insufficient | 6298 | 0.88 (0.79, 0.98) 0.0250 | 0.88 (0.72, 1.08) 0.2308 | 0.96 (0.79, 1.17) 0.6780 |
| sufficient | 4311 | 0.72 (0.64, 0.82) <0.0001 | 0.67 (0.53, 0.86) 0.0015 | 0.75 (0.61, 0.93) 0.0090 |
| **BMI Group** | | | | |
| <25 | 5232 | 0.65 (0.59, 0.71) <0.0001 | 0.75 (0.61, 0.91) 0.0036 | 0.66 (0.56, 0.79) <0.0001 |
| 25-30 | 7562 | 0.78 (0.71, 0.85) <0.0001 | 0.63 (0.54, 0.75) <0.0001 | 0.89 (0.76, 1.05) 0.1678 |
| ≥30 | 9017 | 0.87 (0.80, 0.94) 0.0004 | 0.80 (0.69, 0.93) 0.0027 | 0.98 (0.85, 1.13) 0.7837 |
| **Alcohol consumption** | | | | |
| Never | 3844 | 0.80 (0.73, 0.87) <0.0001 | 0.78 (0.66, 0.91) 0.0020 | 0.80 (0.68, 0.93) 0.0051 |
| Low-to-moderate drinker | 5247 | 0.84 (0.74, 0.94) 0.0039 | 0.72 (0.58, 0.89) 0.0026 | 0.97 (0.77, 1.21) 0.7718 |
| Heavy drinker | 8201 | 0.73 (0.67, 0.79) <0.0001 | 0.66 (0.56, 0.78) <0.0001 | 0.85 (0.73, 0.98) 0.0296 |
| **Smoking** | | | | |
| Never | 11287 | 0.74 (0.67, 0.80) <0.0001 | 0.72 (0.61, 0.85) <0.0001 | 0.80 (0.67, 0.95) 0.0095 |
| Former | 5953 | 0.84 (0.77, 0.91) <0.0001 | 0.81 (0.68, 0.96) 0.0177 | 0.95 (0.82, 1.11) 0.5571 |
| Current | 4874 | 0.77 (0.72, 0.83) <0.0001 | 0.67 (0.58, 0.78) <0.0001 | 0.87 (0.75, 0.99) 0.0401 |
| **CVD** | | | | |
| Yes | 893 | 0.98 (0.85, 1.14) 0.8354 | 1.22 (0.95, 1.58) 0.1198 | 0.82 (0.60, 1.12) 0.2148 |
| No | 21162 | 0.75 (0.71, 0.78) <0.0001 | 0.66 (0.60, 0.73) <0.0001 | 0.86 (0.78, 0.94) 0.0013 |
| **Stroke** | | | | |
| Yes | 839 | 0.78 (0.68, 0.90) 0.0006 | 0.88 (0.70, 1.12) 0.3046 | 0.77 (0.55, 1.06) 0.1125 |
| No | 21262 | 0.77 (0.73, 0.81) <0.0001 | 0.70 (0.63, 0.77) <0.0001 | 0.87 (0.79, 0.95) 0.0024 |
| **Hypertension** | | | | |
| Yes | 9320 | 0.79 (0.74, 0.84) <0.0001 | 0.75 (0.67, 0.84) <0.0001 | 0.89 (0.79, 1.01) 0.0742 |
| No | 12773 | 0.77 (0.72, 0.83) <0.0001 | 0.70 (0.60, 0.82) <0.0001 | 0.83 (0.73, 0.94) 0.0044 |
| **Diabetes** | | | | |
| Yes | 3566 | 0.82 (0.75, 0.90) <0.0001 | 0.78 (0.66, 0.91) 0.0016 | 1.02 (0.83, 1.26) 0.8457 |
| No | 17941 | 0.79 (0.74, 0.84) <0.0001 | 0.73 (0.65, 0.83) <0.0001 | 0.84 (0.76, 0.93) 0.0007 |
| Borderline | 610 | 0.64 (0.47, 0.87) 0.0045 | 0.64 (0.35, 1.18) 0.1507 | 0.54 (0.29, 1.02) 0.0568 |

*(Continued)*

**Table 4.** (Continued)

| Characteristic | No. of participants | All-cause HR (95% CI) p value | CVD HR (95% CI) p value | Cancer HR (95% CI) p value |
|---|---|---|---|---|
| **PIR** | | | | |
| < 1.3 | 5764 | 0.82 (0.75, 0.88) <0.0001 | 0.84 (0.73, 0.97) 0.0181 | 0.83 (0.71, 0.97) 0.0178 |
| 1.3-3.5 | 7175 | 0.80 (0.74, 0.87) <0.0001 | 0.70 (0.60, 0.82) <0.0001 | 0.90 (0.77, 1.05) 0.1705 |
| ≥3.5 | 7356 | 0.81 (0.73, 0.90) 0.0001 | 0.68 (0.54, 0.84) 0.0005 | 0.94 (0.78, 1.13) 0.4992 |

[a] Analyses were adjusted for gender, race, education, martial status, BMI, Physical activity, drinking, smoking, CVD, stroke, hypertension, diabetes, and PIR,except for the stratification variable.

[b] Abbreviations: PIR, ratio of family income to poverty; BMI, body mass index; CVD, cardiovascular disease.

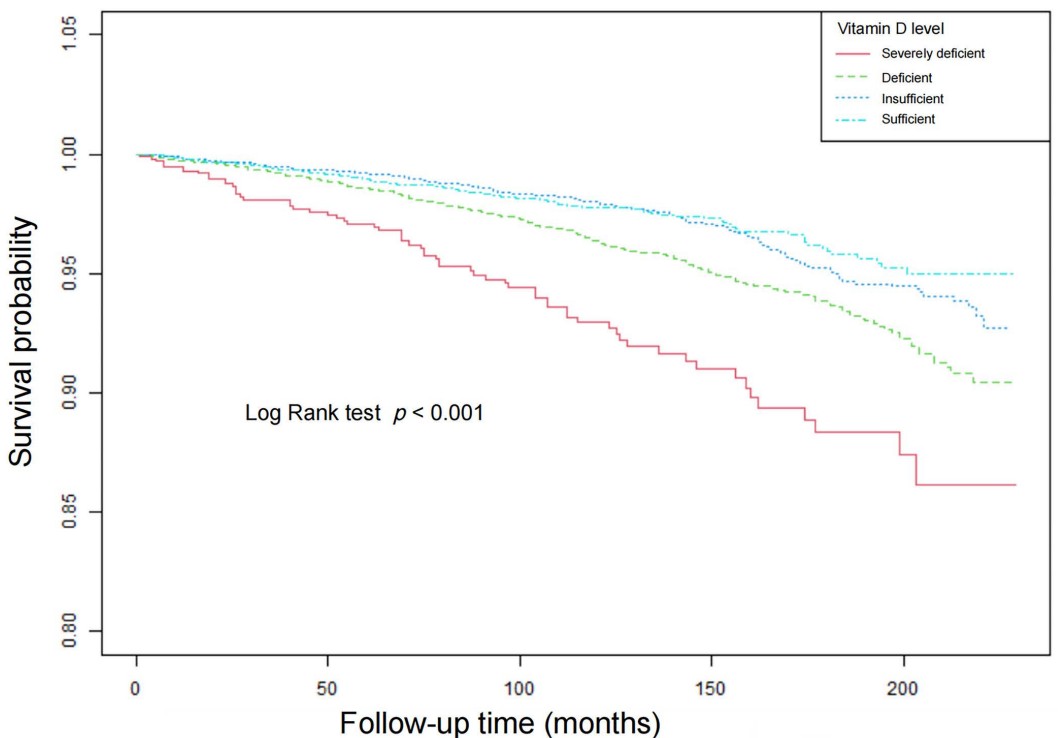

**Fig 3. Survival curves for all-cause mortality by weighted Kaplan-Meier method with Log Rank test.**

## Discussion

This study investigated the relationship between serum 25 (OH) D concentration and all-cause mortality, cardiovascular disease (CVD) mortality, and cancer mortality. Research has found that with an increase in serum 25 (OH) D concentration, all-cause mortality, CVD mortality, and cancer mortality all significantly decrease. Specifically, compared to the group with 25 (OH) D concentrations below 25.00 nmol/L, the group with concentrations ≥ 25.00 nmol/L showed a lower risk of mortality. Further analysis of the dose-response relationship revealed a non-linear relationship between 25 (OH) D concentration and mortality rate. When the concentration was below a specific threshold (such as 54.80 nmol/L, 44.70 nmol/L, and 58.70 nmol/L corresponding to all-cause, CVD, and cancer mortality rates, respectively), an increase in concentration

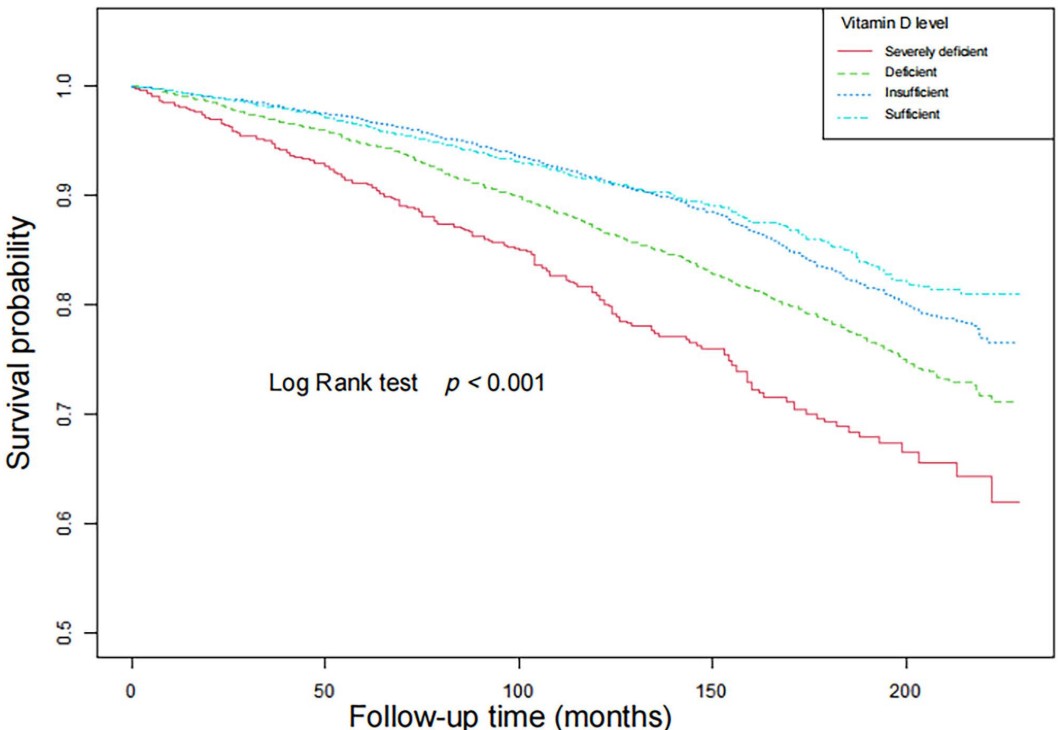

**Fig 4. Survival curves for cardiovascular disease mortality by weighted Kaplan-Meier method with Log Rank test.**

significantly reduced the risk of mortality; And when the concentration exceeds these thresholds, this protective effect is no longer significant. The research results indicate that maintaining an appropriate concentration of 25 (OH) D may play an important role in reducing mortality.

Vitamin D deficiency is a significant public health issue globally [18]. Significant regional and national disparities in serum vitamin D levels have been observed across different populations [19,20]. A community-based epidemiological survey in India analyzed 18 reports, revealing that 50%−94% of the population exhibited vitamin D deficiency, with 51%−91.2% of individuals aged 45–70 years affected [21]. Another study focusing on elderly populations over 60 years in India found that the mean serum vitamin D level was 48.25 nmol/L, with 56.3% demonstrating deficiency (<50 nmol/L), comprising 57.2% of males and 54.2% of females [22]. Studies based on the Chinese population indicate that the prevalence of vitamin D levels <50 nmol/L and <75 nmol/L among adults aged 40 years or older ranges from 60.4% to 72.1% and 78.2% to 90.7%, respectively [23–25]. In contrast, significant regional disparities in vitamin D levels have been observed among middle-aged and older populations in developed countries, with prevalence rates ranging from 4% to 89% for deficiency and 26% to 96% for insufficiency in Europe [26–31]. The data from this study demonstrate that vitamin D deficiency prevalence among US adults aged 40 years or older is significantly lower than in developing countries such as India and China but is similar to that observed in European countries. Factors such as sunlight exposure, latitude, and seasonal variations may influence vitamin D status [18]. Additionally, vitamin D supplements and vitamin D-rich foods are recognized as important cofactors affecting vitamin D levels [18].

This study indicates that there is a non-linear relationship between serum 25 (OH) D levels and all-cause mortality and idiopathic mortality in the middle-aged and elderly population aged 40–70 years. The inflection points corresponding to all-cause, CVD, and cancer mortality are 54.80 nmol/L, 44.70 nmol/L, and 58.70 nmol/L, respectively. However, there is

currently no consensus on the optimal threshold for diagnosing vitamin D deficiency and insufficiency [32]. For example, the American Institute of Medicine defines vitamin D deficiency as serum levels<30 nmol/L, while 30–50 nmol/L is considered vitamin D deficiency [33]. In contrast, the American Endocrine Society defines serum vitamin D levels below 50 nmol/L as deficiency and 50–75 nmol/L as insufficiency [15]. A meta-analysis of 62548 individuals in the general population showed that individuals with serum 25 (OH) D concentrations between 75–87.5 nmol/L had the lowest risk of mortality [34]. A cohort study from the UK Biobank showed that in the general population, the concentration of 25 (OH) D associated with the lowest risk of all-cause mortality was 60 nmol/L [35]. This indicates that the current threshold for defining serum 25 (OHD) adequacy may be too strict for certain populations. Therefore, our findings help determine the optimal threshold diagnosis for specific age groups, which is beneficial for developing health management plans.

The potential mechanism of vitamin D in influencing mortality appears to involve a complex interplay of immune regulation. Specifically, vitamin D plays a role in the differentiation of monocytes and the activation of phagocytic and antibody-dependent macrophage functions. Additionally, it modulates cytokine release and influences lymphocyte activities that produce antibodies, thereby exerting comprehensive control over immune responses [36,37]. High levels of 25-hydroxyvitamin D3 (25(OH)D3) are closely associated with vitamin D-binding protein within monocytes, facilitating their efficient transport in circulation and enhancing their capacity to bind with the vitamin D receptor (VDR) [38,39]. This interaction not only optimizes the availability of 25(OH)D3 in systemic delivery but also triggers the recruitment of autophagy pathways. In these pathways, autophagosomes act as key platforms for antigen presentation, thereby driving the generation of adaptive immune responses [38,39]. Furthermore, White et al. have demonstrated that vitamin D can significantly upregulate the expression of certain antimicrobial peptide genes. Through its interaction with Toll-like receptor (TLR), an immune receptor, vitamin D provides a molecular basis for the protection of the immune system against infectious pathogens [37]. The effect of its role has a potential impact on reducing mortality. Regarding the reduction of CVD mortality risk, 1,25-Dihydroxyvitamin D has been shown to inhibit matrix metalloproteinases (MMPs) and other proteases, thereby preventing the breakdown of elastic proteins in the vascular wall, which helps reduce the risk of arterial stiffness [40]. On the other hand, vitamin D has been proven to inhibit the proliferation of smooth muscle cells, so it may help prevent increased smooth muscle tone or smooth muscle cell hypertrophy, leading to increased arterial stiffness [41]. In addition, our study indicates that compared to the group with severe vitamin D deficiency, the groups with insufficient and sufficient serum vitamin D levels have lower cancer mortality rates.

We speculate that the possible mechanisms underlying the correlation between vitamin D levels and cancer mortality are as follows: Firstly, calcitriol interferes with the growth and aggressiveness of cancer cells by promoting the differentiation of epithelial cells; These cells act as barriers, interfering with the migration of cancer cells through the extracellular matrix [42]. Secondly, calcitriol can also inhibit the proliferation of cancer cells by inducing G1 cell cycle arrest. As the promoter region of the cell cycle inhibitor p21 contains vitamin D response elements, calcitriol can directly regulate the transcription of the p21 gene, thereby exerting anti-tumor cell proliferation activity [43]. Thirdly, active vitamin D inhibits Wnt/β-catenin signaling through multiple pathways [44]. Abnormal activation of the Wnt/β-catenin signaling pathway has been widely observed in various cancers. Its overactivity can promote tumor growth, metastasis and treatment tolerance by maintaining the self-renewal ability of cancer stem cells [44]. For instance, in colorectal cancer, mutations in the APC gene lead to uncontrolled Wnt signaling, causing excessive accumulation of β-catenin and subsequently activating oncogene expression [42]. Vitamin D inhibits the abnormal activation of the Wnt/β-catenin signaling pathway by promoting the binding of VDR to β-catenin and reducing the binding of β-catenin to TCF [42]. Vitamin D upregulates the expression of the CDH1 gene (encoding E-cadherin), promoting E-cadherin to capture β-catenin and preventing it from entering the nucleus to activate downstream oncogenes [45]. Vitamin D also exerts an anti-cancer effect in cancers such as colorectal cancer by increasing the secretion of the Wnt inhibitor DKK-1, further inhibiting the abnormal activation of the Wnt signaling pathway [46]. Fourth, calcitriol exhibits anti-angiogenic effects in both in vitro and in vivo models by inhibiting the growth of tumor-derived endothelial cells (TDECs), reducing endothelial cell germination and morphogenesis, and

lowering the expression of vascular endothelial growth factor (VEGF) and the number of microvessels [47]. Moreover, this effect depends on the vitamin D receptor (VDR) pathway. For instance, the levels of angiogenic factors (such as HIF-1α, VEGF, etc.) increase in VDR knockout mice, while calcitriol can inhibit VEGF-dependent tumor growth in normal mice [48]. Although calcitriol may induce VEGF expression in some cases, its overall anti-tumor effect may stem from the balance of pro-angiogenic and anti-angiogenic factors (such as simultaneous regulation of VEGF and Thrombospondin-1), rather than simply inhibiting angiogenesis [49,50]. In addition, vitamin D can promote the absorption of calcium ions and increase the intracellular calcium ion concentration [51], and an increase in intracellular calcium concentration can trigger apoptosis in various types of cancer cells [52,53]. For instance, breast cancer cells undergo programmed cell death after a continuous inflow of calcium stimulated by 1,25 (OH) 2D3. In contrast, normal breast cells have a reduced susceptibility to calcium-induced apoptosis, which may be due to the enhanced cytoplasmic calcium buffering mechanism. These factors can effectively inhibit the proliferation activity of cancer cells, promote apoptosis of cells, disturb tumor vasculature formation, and modify cell adhesion and migratory ability, thereby subsequently diminishing the metastatic potential of cancer cells [54]. Ultimately reducing the mortality rate of cancer patients.

Research findings indicate that participants from lower socioeconomic groups, characterized by factors such as low income, limited educational attainments, and high body mass index (BMI), exhibit a higher likelihood of vitamin D deficiency or suboptimality. These observations underscore the potential role of social determinants of health in influencing serum vitamin D levels [55,56]. Several plausible explanations account for this association, Differential Dietary Patterns: Lower socioeconomic groups tend to consume fewer whole grains, lean meats, fish, low-fat dairy products, and fresh fruits and vegetables compared to higher socioeconomic groups. Given that these food types are nutrient-rich in vitamin D, this dietary disparity may contribute to the observed differences [57]. Lack of Knowledge: These groups often exhibit limited awareness regarding vitamin D deficiency and appropriate supplementation strategies [58]. Reduced Sunlight Exposure: Individuals from lower socioeconomic backgrounds are frequently exposed to fewer hours of sunlight, which significantly impacts vitamin D synthesis via sunlight exposure [59,60]. This diminished exposure hampers the conversion of 7-dehydrocholesterol into vitamin D3 under UV light action [61]. Despite these challenges, vitamin D deficiency carries multiple health risks. To address these disparities, it is imperative to devise and implement targeted strategies that bridge social divides, thereby reducing health inequalities [62].

This study has three notable strengths. First, it has a large sample size and a long follow-up period, specifically examining the association between serum 25(OH)D concentration and mortality rates in the elderly population. Second, we carefully adjusted for socioeconomic status, dietary habits, lifestyle factors, comorbidities, and other potential confounders, further validating the reliability of our findings. Additionally, the use of standardized methods to determine serum 25(OH)D concentrations in the NHANES database ensures the reliability of our data sources.

However, several methodological limitations warrant attention. First, vitamin D levels are influenced by a wide array of factors, including dietary habits, occupational environments, natural environments (such as UV-B radiation exposure), and personal habits (like sunscreen use and vitamin D supplementation) [63]. Moreover, the varying absorption and utilization efficiencies of different vitamin D supplements [64] were not adequately addressed in this study. Second, seasonal variations in sunlight exposure can significantly affect serum vitamin D levels [65], and the failure to account for seasonal patterns in this study may result in biased estimates of risk factors and outcomes. Additionally, participants' self-reported health status and socioeconomic levels may introduce reporting bias and confounding by socioeconomic status. Consequently, the conclusions of this study are predominantly based on observed associations rather than establishing direct causal relationships.

## Conclusion

This study found that an increase in serum 25 hydroxyvitamin D concentration is significantly associated with a decrease in all-cause mortality, cardiovascular disease mortality, and cancer mortality. Research has shown that there is a

non-linear relationship between the concentration of 25 hydroxyvitamin D and mortality rate. When the concentration is below a specific threshold, an increase in concentration can significantly reduce the risk of death, but when it exceeds the threshold, this relationship has no significant correlation. Further research is imperative to explore the potential causal relationships and underlying mechanisms.

## Supporting information

**S1 File. Raw data.**
(CSV)

**S1 Table. Serum vitamin D levels over the past 30 days among participants grouped by fish consumption levels in the NHANES 2001–2002 survey cycle.**
(DOCX)

## Author contributions

**Conceptualization:** Yaowen Hu, Faliang Gao, Xi Chen, Wenyan Zhao, Xiaopeng He.

**Data curation:** Yaowen Hu, Wei Yang, Jie Zhou.

**Formal analysis:** Yuan Yang, Huibo He, Jie Zhou, Yujie Zhao.

**Investigation:** Yaowen Hu, Wei Yang, Jie Zhou, Yujie Zhao.

**Methodology:** Yaowen Hu, Faliang Gao, Wenyan Zhao.

**Project administration:** Huibo He, Xi Chen, Xiaopeng He.

**Writing – original draft:** Yaowen Hu, Faliang Gao, Yuan Yang, Huibo He, Wenyan Zhao.

**Writing – review & editing:** Xi Chen, Wenyan Zhao, Xiaopeng He.

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
