## [Decision Letter · Decision Letter 0]

PONE-D-25-08106Serum 25(OH)D Levels and Mortality Risk Among Middle-Aged and Elderly Populations in the U.S.: A Prospective Cohort StudyPLOS ONE

Dear Dr. He,

Thank you for submitting your manuscript to PLOS ONE. After careful consideration, we feel that it has merit but does not fully meet PLOS ONE’s publication criteria as it currently stands. Therefore, we invite you to submit a revised version of the manuscript that addresses the points raised during the review process.

We look forward to receiving your revised manuscript.

Kind regards,

Natural Hoi Sing Chu, Ph.D

Academic Editor

PLOS ONE

“This work were supported by Zhejiang Province Health Science and Technology Project (grant no. 2024KY652) and Zhejiang Province Traditional Chinese Medicine Science and Technology Project (grant no. 2025ZL188).”

3. We note that your Data Availability Statement is currently as follows: [All relevant data are within the manuscript and its Supporting Information files]

Additional Editor Comments:

Agreed with reviewers' comment, this topic has been well discussed, please indicated that any novelty addressment of confounding factors like energy or macronutrients intake, etc.

Reviewers' comments:

Reviewer's Responses to Questions

**Comments to the Author**

1. Is the manuscript technically sound, and do the data support the conclusions?

Reviewer #1: Yes

Reviewer #2: Partly

2. Has the statistical analysis been performed appropriately and rigorously? 

Reviewer #1: Yes

Reviewer #2: No

3. Have the authors made all data underlying the findings in their manuscript fully available?

Reviewer #1: Yes

Reviewer #2: Yes

4. Is the manuscript presented in an intelligible fashion and written in standard English?

Reviewer #1: Yes

Reviewer #2: Yes

5. Review Comments to the Author

Reviewer #1: 1- A well written comprehensive manuscript.

2- Relevant Comprehensive statistical test used.

3- Updated references used in the introduction and Discussion both

4- Results according to the study objectives

5- Conclusion aligned with the results

Reviewer #2: - Numerous cohort studies and meta-analyses have already investigated the association between serum 25(OH)D levels and mortality risk in various populations. While the existing literature provides substantial evidence on this topic, it is essential to clarify the necessity and innovation of this study. Specifically, the authors should explicitly state how this study adds to the current body of knowledge.

- The abstract provides a summary of the key findings, including hazard ratios (HRs) for the association between serum 25(OH)D levels and mortality risk. However, to enhance the transparency and scientific rigor of the study, it is important to include the corresponding 95% confidence intervals (CIs) and p-values for the reported HRs.

- While the study presents hazard ratios (HRs) based on quartiles of serum 25(OH)D levels, it would be highly informative to quantify the linear association between continuous serum 25(OH)D levels and mortality risk. Specifically, the authors should consider analyzing and reporting the change in mortality risk per unit increase in serum 25(OH)D levels (e.g., per 10 nmol/L or ng/mL increase). This approach would provide a clearer understanding of the dose-response relationship and allow for more precise interpretation of the results. This analysis should be added to the results section, and its implications discussed in the context of existing evidence.

- The study has done a commendable job of adjusting for several important confounders in the multivariable models. However, two key variables—physical activity and caloric intake—are notably absent from Model 3. Physical activity is a well-known determinant of both vitamin D status (through outdoor activity and sunlight exposure) and mortality risk, while caloric intake reflects overall nutritional status and dietary patterns, which could also influence the observed association. To ensure a more comprehensive adjustment for potential confounding, it is recommended to include both physical activity and caloric intake as additional covariates in Model 3. This adjustment would further strengthen the validity and robustness of the study's conclusions.

- While the results section provides detailed statistical findings, the discussion and conclusion sections should more clearly and transparently summarize the key results to ensure they are accessible to readers. Specifically, the authors should explicitly restate the main findings, such as the magnitude of the association between serum 25(OH)D levels and mortality risk, and highlight any dose-response relationships or subgroup differences.

- The discussion section should explicitly highlight the strengths of the study to provide a balanced perspective and underscore its contributions to the field.

- The study reports an unexpected finding that participants with severe vitamin D deficiency (the lowest quartile of serum 25(OH)D levels) had lower mortality rates compared to other quartiles. This observation contradicts the well-established literature, which generally associates lower vitamin D levels with higher mortality risk. The authors should address this finding in the discussion section and provide potential explanations.

6. PLOS authors have the option to publish the peer review history of their article (what does this mean? ). If published, this will include your full peer review and any attached files.

**Do you want your identity to be public for this peer review?** For information about this choice, including consent withdrawal, please see our Privacy Policy .

Reviewer #1: **Yes: ** Prof. Dr. Uzma Hassan

Reviewer #2: No

---

## [Author Response · Author response to Decision Letter 1]

12 May 2025

Dear Editors and Reviewers:

Thank you for your letter and for the reviewers’ comments concerning our manuscript.Those comments are all valuable and very helpful for revising and improving our paper, as well as the important guiding significance to our researches. We have studied comments carefully and have made correction which we hope meet with approval.

Best regards

---

## [Editor Report · Decision Letter 1]

PONE-D-25-08106R1Serum 25(OH)D Levels and Mortality Risk Among Middle-Aged and Elderly Populations in the U.S.: A Prospective Cohort StudyPLOS ONE

Dear Dr. He,

Thank you for submitting your manuscript to PLOS ONE. After careful consideration, we feel that it has merit but does not fully meet PLOS ONE’s publication criteria as it currently stands. Therefore, we invite you to submit a revised version of the manuscript that addresses the points raised during the review process.

After addressing the reviewers' comments, the manuscript has been improved and now demonstrates novelty, especially showing the intake of total calorie and physical activity from 24 hr recall.

However, I noticed that the total calorie intake among the severely deficient, moderately deficient, insufficient, and sufficient groups is quite similar, even though the p-value is statistically significant.

Additionally, the standard deviation of calorie intake is notably high. Would it be more appropriate to present this data as the median with interquartile range (IQR) using the Kruskal-Wallis test instead?

If available, it would be helpful to elaborate on the consumption of salmon, milk, and its products (which are rich in vitamin D) to better understand how participants could have such low serum vitamin D levels.

Furthermore, could we discuss the possible mechanisms linking vitamin D deficiency to cancer mortality? Is there any possible connection related to the co-absorption of calcium?

We look forward to receiving your revised manuscript.

Kind regards,

Natural Hoi Sing Chu, Ph.D

Academic Editor

PLOS ONE

Journal Requirements:

Reviewers' comments:

NA

---

## [Author Response · Author response to Decision Letter 2]

7 Jul 2025

We sincerely thank the Academic Editor for their constructive feedback. In particular, we extend our gratitude to Dr.Natural Hoi Sing Chu for insightful suggestions, which have significantly strengthened our manuscript.

---

## [Editor Report · Decision Letter 2]

Serum 25(OH)D Levels and Mortality Risk Among Middle-Aged and Elderly Populations in the U.S.: A Prospective Cohort Study

PONE-D-25-08106R2

Dear Dr. He,

We’re pleased to inform you that your manuscript has been judged scientifically suitable for publication and will be formally accepted for publication once it meets all outstanding technical requirements.

Kind regards,

Natural Hoi Sing Chu, Ph.D

Academic Editor

PLOS ONE

Additional Editor Comments (optional):

The authors have addressed the reviewers' and editor's concerns, demonstrating the novel perspective on vitamin D levels and mortality risk.
---

## [Editor Report · Acceptance letter]

PONE-D-25-08106R2

PLOS ONE

Dear Dr. He,

I'm pleased to inform you that your manuscript has been deemed suitable for publication in PLOS ONE. Congratulations! Your manuscript is now being handed over to our production team.

Kind regards,

on behalf of

Dr. Natural Hoi Sing Chu

Academic Editor

PLOS ONE